# Learning Elementary Cellular Automata with Transformers

**Mikhail Burtsev**
London Institute for Mathematical Sciences
Royal Institution, UK
mb@lims.ac.uk

## Abstract

Large Language Models demonstrate remarkable mathematical capabilities but at the same time struggle with abstract reasoning and planning. In this study, we explore whether Transformers can learn to abstract and generalize the rules governing Elementary Cellular Automata. By training Transformers on state sequences generated with random initial conditions and local rules, we show that they can generalize across different Boolean functions of fixed arity, effectively abstracting the underlying rules. While the models achieve high accuracy in next-state prediction, their performance declines sharply in multi-step planning tasks without intermediate context. Our analysis reveals that including future states or rule prediction in the training loss enhances the models' ability to form internal representations of the rules, leading to improved performance in longer planning horizons and autoregressive generation. Furthermore, we confirm that increasing the model's depth plays a crucial role in extended sequential computations required for complex reasoning tasks. This highlights the potential to improve LLM with inclusion of longer horizons in loss function, as well as incorporating recurrence and adaptive computation time for dynamic control of model depth.

## 1 Introduction

Large Language Models (LLMs) have become valuable tools in mathematics, demonstrating impressive capabilities in problem-solving and reasoning tasks. Notably, OpenAI's o1 model achieved a ranking among the top 500 students in the US in a qualifier for the USA Math Olympiad (AIME) [1]. Despite these successes, LLMs still face challenges in reasoning [2–6] and planning [7], particularly when required to infer and apply underlying rules from data.

These observations raise a question: Do the limitations of LLMs in reasoning stem from the nature of their training data, the procedures employed during training, or inherent architectural constraints?

Transformers [8], which form the backbone of many modern LLMs, are universal approximators [9–12] and theoretically capable of simulating Turing machines using intermediate computational steps, making them Turing complete and, by extension, capable of formal reasoning [13–16]. Earlier studies found that transformers can be trained to perform symbolic integration and solving differential equations [17], as well as symbolic regression [18, 19].

In our study, we continue the line of research exploring the trainability of Transformers for mathematical reasoning tasks by focusing on Elementary Cellular Automata (ECAs). The simplicity and clarity of this "toy" problem make it an ideal testbed for assessing the ability of Transformers to abstract and apply logical rules. By demonstrating that Transformers can learn and generalize Boolean functions of fixed arity inherent in ECAs, we aim to evaluate their ability to infer, generalize, and apply logical rules solely from observed data, without relying on memorization.

38th Conference on Neural Information Processing Systems (NeurIPS 2024).

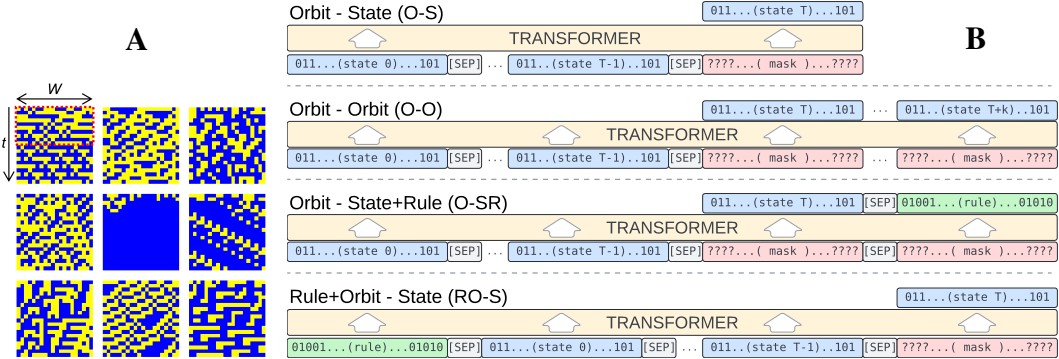

Figure 1: **Learning Elementary Cellular Automata (ECA) with Transformers. A.** Examples of training samples. Orbit of ECA is a sequence of binary strings of size $W = 20$. First $k = 10$ states marked by red rectangle encode Transformer input. **B.** Given a part of the orbit Transformer with full attention learns to predict the next state (O-S), the next few steps (O-O), the next state and a rule (O-SR), or predict the next state given a rule and an orbit (RO-S).

## 2 Methods

An *Elementary Cellular Automaton (ECA)* is a one-dimensional, dynamical system in which space and time are discrete. Let $r \in \mathbb{N} : r \geq 1$ be the *neighbourhood radius* in space represented by a regular lattice of $W \in \mathbb{N} : W \geq 2r + 1$ identical, locally-interconnected *cells* with a binary state space, $\mathbb{S} = \{0, 1\}$. The ECA's *global state*, $x \in \mathbb{S}^W$, is a lattice configuration specified by the values of all the states of all cells in the lattice at a given time. This state evolves deterministically in synchronous, discrete time steps according to a *global map* $g_\rho : \mathbb{S}^W \to \mathbb{S}^W$ defined by a *local rule* $\rho : \mathbb{S}^{2r+1} \to \mathbb{S}$, so $[g_\rho(x)]_w = \rho(x_{w-r}, \ldots, x_w, \ldots, x_{w+r})$. The sequence of states an ECA passes through during its *space–time evolution*, $\mathcal{O}^T(x) = [x, g_\rho(x), g_\rho(g_\rho(x)), \ldots, g_\rho^{oT-1}(x)]$, defines its *trajectory* or *orbit* from an *initial condition* (configuration) $x$ for $T \in \mathbb{N} : T \geq 1$. Examples of ECA orbits are visualized on the figure 1A.

Let consider four modifications of learning tasks designed to evaluate different aspects of predictive modeling and rule inference in ECAs (see Fig. 1B).

*Orbit-State (O-S),* given an orbit $\mathcal{O}^T(x) = [x^{(0)}, x^{(1)}, \ldots, x^{(T-1)}]$ where $x^{(0)} \in \mathbb{S}^W$, the objective is to predict the state $x^{(T)}$ at time $T$.
*Orbit-Orbit (O-O),* given an orbit $\mathcal{O}^k(x)$ for some $k < T$ predict the subsequent states up to time $T$, generating $\mathcal{O}_k^T(x) = [x^{(k)}, \ldots, x^{(T)}]$.
*Orbit-State and Rule (O-SR),* given an orbit $\mathcal{O}^T(x)$ the objective is to predict the state $x^{(T)}$ at time $T$ and the local rule $\rho$.
*Rule and Orbit-State (RO-S),* given an orbit $\mathcal{O}^T(x)$ and the local rule $\rho$ the objective is to predict the state $x^{(T)}$ at time $T$.

Our base model is Transformer encoder with full self-attention. It has 4 layers and 8 heads with $d_{model} = 512$. The input vocabulary of the model consists of tokens: [0], [1], [SEP], and [M].The states and the local rule $\rho$ are encoded as binary strings. The model receives the orbit as a sequence of bits representing consecutive states separated by the [SEP] tokens. For the prediction of future states or the inference of the local rule, the end section of the input sequence is filled with mask tokens [M] corresponding to the positions of the unknown elements.

We generated a dataset with the CellPyLib [20] [1] for fixed lattice size $W = 20$ and neighborhood radius $r = 2$. This configuration results in a total of $2^{2^{2r+1}} \approx 4.3 \times 10^9$ possible boolean functions defining local rules. For each sample in the dataset, both the initial state and the local rule $\rho$ were generated randomly. We then computed the orbit for $T = 20$ time steps using these parameters. The training dataset consists of $9.5 \times 10^5$ and the test of $10^5$ samples. Importantly, the local rules included in the test set are exclusive and not present in the training set. This separation ensures that the model's

---

[1]Dataset and a source code for experiments are available at `https://github.com/burtsev/TransformerECA`.

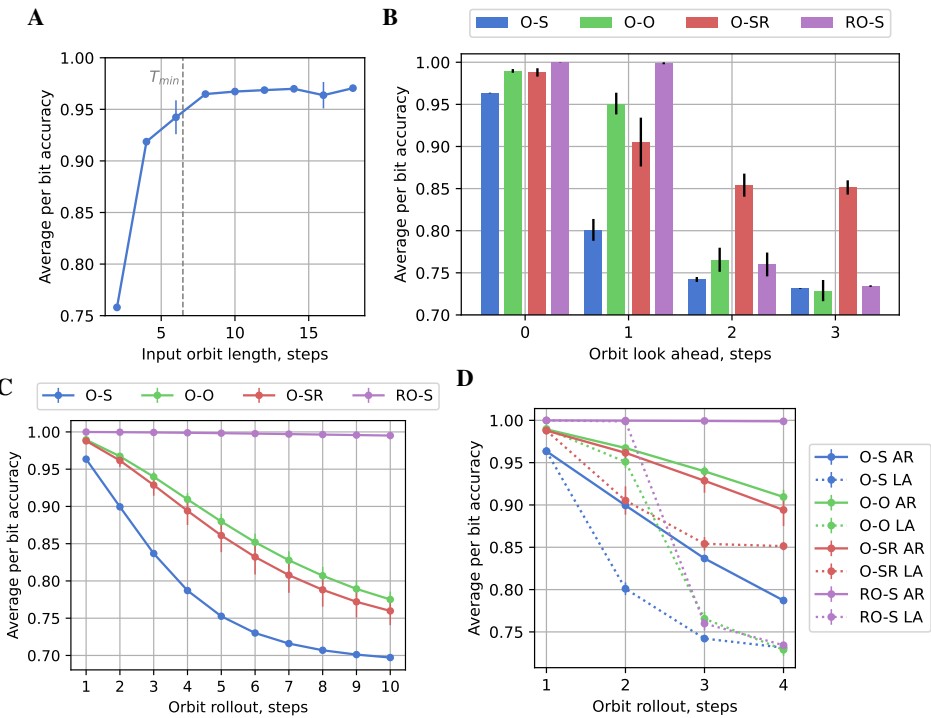

Figure 2: **Transformer learns to predict the next state of ECA but struggles to plan ahead.**
**A.** Accuracy of the next state prediction for ECA orbit with size 20 generated by Boolean function of 5 arguments for different input state lengths. **B.** Planning accuracy for different training settings (see the main text for details). **C.** Accuracy for autoregressive generation of ECA orbit. **D.** Comparison of autoregressive (AR) and look ahead (LA) predictions.

performance reflects its ability to generalize to unseen rules, rather than simply memorizing the training data. To measure the quality of the model's predictions, we use per-bit accuracy averaged over free runs, which calculates the proportion of bits correctly predicted in the output sequences.

# 3 Results and Discussion

We first assess whether the samples in our dataset provide sufficient information for the model to learn the underlying dynamics successfully. The minimal number of time steps $T_{\min}$ needed to recover the local rule $\rho$ from the observed orbit can be estimated analogous to the coupon collector's problem $T_{\min} = 2^{2r+1}(\ln 2^{2r+1} + \gamma)/W \approx 6.47$, where $\gamma$ is the Euler-Mascheroni constant ($\gamma \approx 0.5772$). Training of the model for the next state prediction (O-S task) confirms this theoretical estimation by showing that the accuracy plateaus after 8 time steps (Fig. 2A). Therefore, we choose to use input orbits of 10 steps $\mathcal{O}^{10}$ as our main setting.

Successful learning of the O-S task demonstrate that the Transformer model is capable of generalizing not only over initial conditions for a particular function — commonly the focus in studies of transformer trainability in CA domain [21–35] — but also across different Boolean functions of fixed arity (5 in our case). This indicates that the model has learned to abstract a class of rules.

Next, we investigated the Transformer's ability to plan ahead by predicting future states beyond the immediate next state. Specifically, we trained the model to predict the state at time $x^{(T+k)}$ for look-ahead steps $k \in \{1, 2, 3\}$. As presented in Figure 2B, this task proved to be significantly more challenging. While the average accuracy for next-state prediction (O-S task) was 0.96, it dropped to 0.80 for $k = 1$ and fell below 0.75 for $k = 2$ and $k = 3$.

To determine whether this decline was due to the Transformer's architecture or the training objective, we explored whether accuracy could be improved by training the model to predict intermediate steps. This approach is analogous to the "chain-of-thought" method [36] used in large language models for

in-context learning. We employed the Orbit-Orbit (O-O) task, training the model to predict the next four states in parallel. The results, also shown in Figure 2B, indicate that the model retains predictive abilities when skipping one step but struggles with skipping two or three steps.

Moreover, if we hypothesize that during training the Transformer learns to generate an internal representation of the local rule $\rho$ and then applies it to predict the next state, augmenting the training task with explicit rule prediction might help the model form this internal rule representation more effectively. To test this hypothesis, we employed the Orbit-State and Rule (O-SR) training for planning ahead without intermediate steps. For next-state prediction, the O-SR model achieved performance comparable to the O-O model, indicating that predicting future states and inferring the rule have similar effects on the model's learning process. As presented in Figure 2B, for $k = 1$, the O-SR model attained an accuracy of 0.91 compared to 0.95 for the O-O model. However, for $k = 2$ and $k = 3$, the O-SR model outperformed the O-O model with accuracies of approximately 0.85 versus 0.75, respectively.

These results suggest that learning to store a hidden representation of intermediate states (as in the O-O training) is easier for the model but harder to generalize over multiple time steps. In contrast, developing a hidden representation of the underlying rule (as in the O-SR training) is more challenging initially but facilitates better generalization to longer planning horizons. This implies that explicitly encouraging the model to infer the generating rule can enhance its ability to make longer-term predictions by reinforcing the internalization of the system's dynamics.

Finally, we explored the scenario where the local rule $\rho$ is explicitly provided to the model, corresponding to the Rule and Orbit-State (RO-S) task. Intuitively, this should be the easiest task for the Transformer, as it eliminates the need to infer the rule from the observed data. As shown in Figure 2B, the Transformer indeed learns to apply the given rule for next-state prediction with near-perfect accuracy for $k = 0$ and $k = 1$. Surprisingly, however, the performance for look-ahead steps $k = 2$ and 3 drops to approximately 0.75, similar to the O-O training scenario where the rule is not provided.

This unexpected decline hints that even with explicit access to the rule, the model struggles to predict future states beyond the two next steps. We hypothesize that this limitation arises from difficulties in learning effective hidden representations of the intermediate states required for multi-step predictions. Despite having the rule, the Transformer may not adequately capture and propagate the necessary state information over multiple time steps without explicit intermediate context.

Additionally, we evaluated the performance of the four models — trained under the O-S, O-O, O-SR, and RO-S tasks — when used to generate continuations of the input orbit $\mathcal{O}^{10}$ up to $\mathcal{O}^{20}$ by predicting each subsequent state autoregressively (see Figure 2C). As expected, the success of these models in this task correlates with their next-token prediction accuracy (refer to the first group of bars in Figure 2B). The RO-S model exhibits the best performance, followed closely by the O-O and O-SR models, while the O-S model shows significantly weaker performance.

When comparing autoregressive generation (AR) for rollouts of 2, 3, and 4 steps to planning performance for the same look-ahead steps (LA) (Figure 2D), we observe that all models perform better in the autoregressive inference mode. This suggests that the models are more adept at short-term state-by-state prediction than at planning multiple steps ahead without intermediate context. The superior performance in the autoregressive mode highlights the models' reliance on immediate past states to inform their predictions.

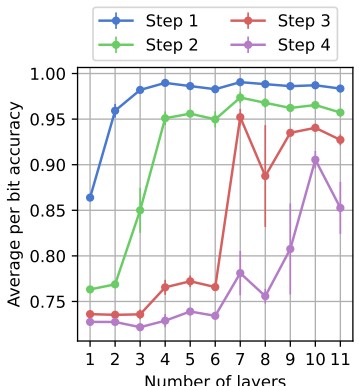

Figure 3: **Adding layers improves prediction of ECA orbit.** Accuracy of O-O training for different number of layers.

Our planning results (Figure 2B) demonstrate a sharp decline in performance across all training methods starting from a look-ahead of 2 steps. This decline may be due to the Transformer's inability to store more than one ECA state in its hidden representations or because the task requires sequential application of the local rule equal to the planning horizon. The O-O training method explicitly provides memory

for intermediate states through tokens and associated hidden vectors, effectively ruling out the first explanation.

The alternative explanation points to the limitations of sequential computation within the Transformer architecture, which is bounded by the number of layers. To test this, we examined how the ability to predict future states scales with network depth. Figure 3A shows that the O-O model begins to accurately predict the next state with 2 layers. Predicting two steps ahead requires at least 4 layers, three steps necessitate 7 layers, and four steps demand 10 layers. These results suggest that each additional planning step requires more computational layers, highlighting a direct relationship between the Transformer's depth and its capacity to model longer sequences of ECA dynamics.

## 4    Conclusions

In this study, we have demonstrated that Transformer models possess the ability to learn and generalize the underlying dynamics of Elementary Cellular Automata. By designing specific tasks and training regimes, we showed that Transformers can abstract the governing rules and predict future states with notable accuracy. However, their performance diminishes when required to plan multiple steps ahead without intermediate context, highlighting limitations in storing and propagating state information over longer sequences. Our analysis reveals that including future states or rule prediction in the training loss improves the performance of next-state prediction and, as a result, enhances the overall performance in autoregressive generation. This finding might guide training strategies for LLMs to improve their perplexity and reasoning capabilities. We also confirmed that the model's depth plays an important role in extended sequential computations required for complex reasoning; thus, recurrence and adaptive computation time are promising directions for dynamic control of the model depth. These insights contribute to a deeper understanding of how neural networks can abstract rules and point toward future research directions in improving their planning and generalization skills.

## Acknowledgments

Computational grant for this work was provided by Nebius AI Cloud.

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
