# OpenReview forum: "Learning Elementary Cellular Automata with Transformers"
_NeurIPS.cc/2024/Workshop/MATH-AI — MATH-AI 24_

### Official Review · Reviewer_5ovd · 2024-10-05
**A comprehensive work, but not novel enough.**

**Rating:** 4
**Confidence:** 4

**Review:**

This paper explores the potential of Transformers to learn and generalize the rules governing Elementary Cellular Automata (ECAs). By training on sequences generated with random initial conditions and local rules, the study examines whether Transformers can abstract from specific instances to a generalized understanding of Boolean functions. Despite achieving high accuracy in next-state prediction, the models face significant challenges in multi-step planning tasks, which highlights the limitations in their capacity for abstract reasoning and dynamic problem-solving. The paper suggests improvements in model architecture, such as increasing depth and incorporating dynamic elements, to enhance performance on complex reasoning tasks.

Strengths:

1.	The study innovatively uses ECAs as a testbed to examine the generalization capabilities of Transformers across different Boolean functions, highlighting a novel way of evaluating deep learning models' abstract reasoning capabilities.

2.	The paper provides a thorough examination of the models' performance, including different setups such as next-state prediction and multi-step planning. This comprehensive approach allows for a detailed understanding of the models' strengths and limitations.

Weaknesses:

1.	While ECAs provide a clear and controlled environment for testing, they might not capture the complexity of real-world scenarios that Transformers are expected to handle, potentially limiting the broader applicability of the results.

2.	The suggestion to increase model depth might lead to more resource-intensive models, which could be impractical for certain applications due to increased computational and energy costs.

3.	Although the paper aims to teach models to generalize, the specific focus on ECAs and Boolean functions might lead the models to overfit to these types of tasks, potentially reducing their effectiveness on more varied real-world problems.

4. Limited Novelty, learning a Turing Machine has been studied by previous work[1-3], this is a more simple case.



[1] Wei C, Chen Y, Ma T. Statistically meaningful approximation: a case study on approximating turing machines with transformers[J]. Advances in Neural Information Processing Systems, 2022, 35: 12071-12083.

[2] Ryoo M S, Gopalakrishnan K, Kahatapitiya K, et al. Token turing machines[C]//Proceedings of the IEEE/CVF Conference on Computer Vision and Pattern Recognition. 2023: 19070-19081.

[3] Pérez J, Barceló P, Marinkovic J. Attention is turing-complete[J]. Journal of Machine Learning Research, 2021, 22(75): 1-35.

---

### Official Review · Reviewer_WTzu · 2024-10-06
**This paper studies whether Transformer models can learn Elementary Cellular Automata. The authors find that the model's performance drops quickly in multi-step planning tasks. The paper also propose including future states and developing a hidden rule to facilitate better in long planning horizons.**

**Rating:** 6
**Confidence:** 3

**Review:**

This paper explores whether Transformer models can learn and generalize the rules of Elementary Cellular Automata (ECA). By training Transformers on sequences of states, the authors found that the models can generalize across Boolean functions and effectively abstracting the underlying rules.  Although the models exhibit high accuracy in predicting the next state, the paper finds that their performance drops sharply in multi-step planning tasks. By including future states or hidden rule predictions in the training loss, the models' ability to form internal representations of the rules is enhanced, leading to improvements in longer planning horizons.

Strengths:

1. The empirical analysis of the algorithm is interesting. The author compares different methods (O-S, O-O, O-SR, etc) to illustrate the
 relationship between accuracy and planning steps. Moreover, The authors also study the role of depth in modeling long sequences of ECA dynamics.

2.  The research results provide guidance for improving training strategies of large language models when facing long term planning tasks, such as including longer horizon hidden rules in the loss function.

Questions and Suggestions: I'm not an expert in Transformer Model so please forgive me if my questions are naive.

1: May one ask in your setting, how will the lattice and radius sizes influence the results? Thanks!

2: Are there any theoretical guarantees in learning theory between transformer’s ability of planning and the number of layers?

3: Just out of curiousity, have you ever considered more complicated learning tasks rather than Elementary Cellular Automata? In other words, do you expect that your finding could be generalized to other kinds of multistep learning tasks?

---

### Official Review · Reviewer_QnYL · 2024-10-06

**Rating:** 8
**Confidence:** 4

**Review:**

This paper studies the learning of elementary cellular automata (ECA) with Transformers. More precisely, the authors evaluate their models on two tasks: predicting the next state of ECAs and planning ahead i.e. predicting the steps ahead from the observed ECA. They consider different training tasks: Orbit-State (O-S), Orbit-Orbit (O-O), Orbit-State and Rule (O-SR) and Rule and Orbit-State (RO-S) and report the performance of the model on the evaluation tasks. They basically observe that when training with RO-S, they got the best performance on predicting the next state. However, the model then is very bad at planning. They find that the best model for planning is O-SR and hypothesize that forcing the model to learn to generate the rule is the reason for these benefits. Lastly, they do a study with respect to the depth of the model and show that deeper networks are better at planning.

Overall, I really liked this paper and this is why I advocate for acceptance of this paper. It is well explained, studies a problem that has not been studied in Transformers and math, and the findings are interesting.

A few suggestions that may be interesting to the authors:

- First, maybe I am wrong but is this ECA learning analogous to learning automata? There are some works on learning automata with Transformers and I think it would be good to cite them / reflect your work with respect to these works.

- I am not familiar at all with ECAs. Maybe it would be useful to add a small example / drawing in the Methods section to give concrete examples of an orbit and a local rule? This would greatly help the understanding.

- An interesting method to try is a more explicit comparison with Chain-of-Thought (CoT). The authors claim that O-O is the same as CoT but maybe we can make a more explicit CoT? For instance, putting the local rule output next to the intermediate steps to guess. The reason I am insisting on CoTs is because I really believe that if a model learns with CoT, it should be able to do planning. Not entirely related to this but [2] show that with CoT, they can achieve reasonable length generalization on any task.

- I believe that there are other interesting tricks to maybe achieve better planning (inspired from the automata and Transformers literature):

* data mixing: here, it means that i would create a dataset made of O-S tasks but for some samples, the state to predict is x^T, others we need to predict x^T+1, or x^T+2 etc. This is also another implicit way to enforce the model to infer the rule.

* recurrent architectures: how would LSTMs work on this task? I believe that they should do great even when trained on O-S? this would be interesting to see. Other variants would also be worth studying like Universal Transformers [3] or Looped Transformers [4].



[1] Liu, B., Ash, J.T., Goel, S., Krishnamurthy, A. and Zhang, C., 2022. Transformers learn shortcuts to automata. arXiv preprint arXiv:2210.10749.

[2] Hou, K., Brandfonbrener, D., Kakade, S., Jelassi, S. and Malach, E., 2024. Universal Length Generalization with Turing Programs.

[3] Dehghani, M., Gouws, S., Vinyals, O., Uszkoreit, J. and Kaiser, Ł., 2018. Universal transformers.

[4] Giannou, A., Rajput, S., Sohn, J.Y., Lee, K., Lee, J.D. and Papailiopoulos, D., 2023, July. Looped transformers as programmable computers.

---

### Official Review · Reviewer_WJnR · 2024-10-07
**Good analysis on the learning dynamics and capabilities of Transformers**

**Rating:** 7
**Confidence:** 4

**Review:**

**Summary**

In this paper the authors have tried to study the ability of Transformers (and in turn LLMs) to infer and apply abstract rules for long-term planning tasks. They use Elementary Cellular Automata as a “toy example” to test the trainability of Transformers for mathematical and reasoning tasks.

**Strengths**

The different experiments conducted and the related plots were well explained. The authors did a nice analysis of the internal learning dynamics and learning capabilities of Transformers. The work tries to hypothesize and conduct some experiments to prove that the Transformers find it difficult to learn effective hidden representations of the intermediate states required for multi-step predictions. The Transformers get relatively more capable at short-term state-by-state prediction than at planning multiple steps ahead without intermediate context.
The authors conduct experiments to claim that Transformers can be made better at long-horizon multi-step predictions by making them learn the dynamics explicitly by making them predict the underlying rules.
This is aligned to learning some world representations or world dynamics in order to be able to do longer horizon state predictions and planning.
This also leads the authors to conduct experiments that try to explain the relationship between capability and capacity of the Transformers. The experiments try to explain that storing additional context for additional planning steps requires more computational layers and can be done by increasing the depth of the model.

These results might help in designing training pipelines for larger architectures and models for long horizon planning and reasoning.

**Weaknesses**

The paper can mention some of the training details along with some hyper params (although the authors mention sharing the code upon acceptance).

**Limitations**

It would have been nice to have an additional supporting test-bed/toy example to make the claims more effective and broadly applicable.
Was there enough data provided to test the generalization capability of the models?

---

### Decision · Program_Chairs · 2024-10-09

**Decision:**

Accept

**Comment:**

The reviewers are generally positive about the work and excited about the insights that it uncovers about the learning processes of transformers. We recommend that the authors include a more comprehensive comparison with related work in their revision.